# Relationship between Performance and Inter-Limb Asymmetries Using Flywheel Resistance Device in Elite Youth Female Basketball Players

**DOI:** 10.3390/biology11060812

**Published:** 2022-05-25

**Authors:** Azahara Fort-Vanmeerhaeghe, Ariadna Benet-Vigo, Alicia Montalvo, Adrià Arboix, Bernat Buscà, Jordi Arboix-Alió

**Affiliations:** 1Department of Sports Science, Ramon Llull University, FPCEE Blanquerna, 08025 Barcelona, Spain; azaharafv@blanquerna.url.edu (A.F.-V.); ariadnabv@blanquerna.url.edu (A.B.-V.); bernatbs@blanquerna.url.edu (B.B.); 2Segle XXI Female Basketball Team, Catalan Federation of Basketball, 08950 Esplugues de Llobregat, Spain; 3School of Health Sciences, Ramon Llull University, FCS Blanquerna, 08025 Barcelona, Spain; 4College of Health Solutions, Arizona State University, Phoenix, AZ 85004, USA; alicia.monta@gmail.com; 5Cerebrovascular Division, Department of Neurology, Hospital Universitari del Sagrat Cor, Universitat de Barcelona, 08029 Barcelona, Spain; aarboix@quironsalud.es

**Keywords:** imbalances, iso-inertial, change of direction, jumping, symmetry

## Abstract

**Simple Summary:**

Inter-limb asymmetry is defined as the difference in performance or function between limbs. Inter-limb asymmetries have become the focus of recent research, with many studies describing differences in performance on the right and left sides. Most of these studies have reported asymmetry values during unilateral jumps or COD tests, but few have used flywheel resistance (iso-inertial) devices. Our study quantified inter-limb asymmetries using a variety of methods and compared asymmetry with physical performance in a sample of elite youth female basketball players. The results of this set of tests indicated that mean asymmetry magnitudes greatly varied among all metrics and showed different directionality, thus highlighting the lack of consistency across the tests and the task-specific nature of inter-limb asymmetry. Existing recommendations note that a battery of tests is needed to gain a holistic picture of inter-limb asymmetries (such as jumps, changes of direction, or power-specific skills). Identifying inter-limb asymmetries could help practitioners determine the workload ratios for each limb during individual training sessions. Moreover, the use of flywheel resistance devices could be included in the battery of tests for the detection of inter-limb asymmetries.

**Abstract:**

The purposes of this study were to quantify inter-limb asymmetries from unilateral jumps, change of direction (COD) speed, and flywheel resistance skill tests and to examine their relationship with physical performance in a sample of elite youth female basketball players. Eleven female basketball players (age = 17.56 ± 0.60 year; body mass = 75.13 ± 12.37 kg; height = 1.83 ± 0.08 m; BMI = 22.42 ± 2.28; sports experience = 6.31 ± 1.73 year; years post-peak height velocity = 4.79 ± 0.68 year) performed a battery of fitness tests in the post-season consisting of the Single Leg Countermovement Jump in vertical (SLCJ-V), horizontal (SLCJ-H), and lateral (SLCJ-L) directions, 135° and 90° COD tests, and four skills (acceleration step, deceleration step, sidestep, and crossover step) with an flywheel resistance device. The results showed significant differences between the higher performing and lower performing limbs across all tasks (*p* < 0.05). The mean asymmetry index values ranged from 1.26% (COD 135°) to 11.75% (SLC-V). Inter-limb asymmetries were greatest during the flywheel resistance skills. Spearman’s correlations (*ρ*) for all tests were only significant for inter-limb asymmetries during the sidestep test and reduced performance in SLCJ-L (*ρ* = −0.61; *p* = 0.046) and all COD deficits (*ρ* range = −0.72 to −0.81). The findings of the present study showed that inter-limb asymmetries are task-specific in female youth basketball players and suggest that the use of flywheel devices can be included in the battery of tests to detect inter-limb asymmetry.

## 1. Introduction

Basketball is an intermittent team sport characterized by different unilateral, high-intensity actions, such as jumping and changes of direction (COD), which are linked to decisive moments during the game [1,2]. Basketball players frequently perform rapid decelerations, CODs, and sprints to create space or to react to an opponent or the ball. As such, coaches and researchers alike are always striving for more efficient and effective techniques to optimize basketball players’ skills [3].

Because these high-intensity unilateral actions are risky, reducing the injury rate is an important objective of basketball training programs [4]. Inter-limb strength and coordination asymmetries are considered injury risk factors due to the high neuromuscular demands during sport-specific actions [5]. Thus, assessing inter-limb asymmetries during jumping and COD provides valuable information to act on to reduce injury risk in basketball players [6,7,8]. Moreover, the use of inter-limb asymmetry to compare the performance of one limb to the other has been used in the field of rehabilitation to determine when an athlete can return to sport following injury [6]. Although controversial, previous research suggests a 10–15% threshold of inter-limb asymmetry in strength/power and unilateral jumping actions as ‘normal’ physiological variability in team sport athletes [9,10,11]. Relatedly, recent studies agree that direction (right or left) is more accurate than magnitude for detecting inter-limb asymmetries in healthy subjects [12]. Various studies have shown a lack of consistency in the direction of asymmetry during different skills, such as COD and jumping [12,13].

Recently, one of the topics that has garnered much interest in the scientific community is the association between inter-limb asymmetries and physical performance in healthy athletes. Some evidence in team sport athletes shows a relationship between inter-limb asymmetries and decreased jump height [14], decreased acceleration [15], lower sprint performance [16,17], and lower COD performance [18]. However, the true effect of inter-limb asymmetry on physical performance remains unclear [6].

Most of the aforementioned studies have reported asymmetry values during unilateral jumps or COD tests. While asymmetry assessments for these tasks are common, strength and power are less frequently assessed for asymmetry in field-based testing [19]. Flywheel resistance (iso-inertial) devices have been reported in recent studies to be useful tools for assessing asymmetry [20,21]. These devices involve force production during both phases of muscle contraction (concentric and eccentric). This makes flywheel exercises sport-specific, as most sports movements occur through a combination of concentric and eccentric actions, including the stretch-shortening cycle [22]. Because flywheel resistance training transfers to sport [23], it is being increasingly used as a training method in team sports to reduce injury risk and improve performance [24]. In a recent study, Madruga-Parera, Bishop, Beato, et al. [20] quantified the asymmetries in flywheel resistance performance in youth male handball players and found that larger asymmetries were associated with reduced COD tests and sprint performance. They also reported that asymmetries during the concentric phase of flywheel resistance skills correlated with reduced performance in 20 m sprint and COD tests. In addition, Raya-González et al. [21] reported that flywheel resistance tests appeared to be highly sensitive to detecting asymmetries. However, they did not find a significant relationship between flywheel resistance asymmetries and sprint or jumping abilities. Given the conflicting findings in the literature, further studies are needed to determine the relationship between inter-limb asymmetries and athletic performance using flywheel resistance devices. Furthermore, and to the best of the authors’ knowledge, such studies on female athletes do not currently exist.

Therefore, the aims of this study were twofold: (1) to determine the magnitude and directionality of inter-limb asymmetries using jumping, COD, and flywheel resistance skills tests and (2) to study the relationship of asymmetries to physical performance in youth elite female basketball players.

## 2. Materials and Methods

The current study aimed to assess inter-limb asymmetry in elite youth female basketball players using ten tests. These tests measured unilateral jump distance (in the horizontal, vertical, and lateral directions), 90° and 135° COD time, 10-m sprint time, and mechanical power output (flywheel resistance) in different sport-specific skills (acceleration step, deceleration step, crossover step, and sidestep).

### 2.1. Participants

A total of eleven elite youth female basketball players were included in this study: age (17.56 ± 0.60 year), body mass (75.13 ± 12.37 kg), height (1.83 ± 0.08 m), body mass index (22.42 ± 2.28 kg (m^2^)**^−^**^1^), sports experience (6.31 ± 1.73 year), and years post peak height velocity (4.79 ± 0.68 year). Biological maturation was calculated using an equation from Mirwald et al. [25]. All participants were involved in a four-year talent development program at the time of the study. They usually had 7–9 training sessions per week plus one game during the weekend. Before the study started, the participants and their parents received detailed verbal and written information about the possible risks and discomfort associated with doing the tests. Written informed consent was obtained from the participants and from parents/tutors. This study was approved by the Ramon Llull University Ethics Committee (1718007D) and conformed to the recommendations of the Declaration of Helsinki.

### 2.2. Design and Procedures

One week before data collection, the participants were familiarized with all tests and procedures. Participants were tested on five separate days, each separated by 48 h, and the training was always on the same schedule to control the circadian rhythms. In the first four testing days, the following flywheel skills data were collected in order: (1) acceleration step, (2) deceleration step, (3) sidestep, and (4) crossover step. Day 5 consisted of unilateral jump, 10 m linear sprint, and COD tests. During testing days, all participants completed the same standardized warm-up consisting of 5 min of cardiovascular exercise (RPE 5–6), 6 min of multidirectional displacements, 4 min of dynamic stretching exercises (e.g., walking lunges and side steps high knee lifts), and 3 min of maximal and progressive intensity displacements including changes of direction, jumps, and acceleration/deceleration movements. During the warm-up, there was supervision by a qualified strength and conditioning coach, and consistent feedback was provided throughout all tests to ensure a proper technique. On completion, 3 practice trials were provided for each test where participants were instructed to perform them at 70, 85, and 100% of their perceived maximal effort. Two minutes of rest were given between the last practice trial and the start of the first test. Both the order of the tests and the participants were randomized using the “true random number generator” program.

#### 2.2.1. Linear Sprint

Linear sprint speed was evaluated using a 10-m sprint test. The start and finish lines were clearly marked with cones. The time was registered using electronic light gates (PME10D Velleman, Velleman, Inc., Gavere, Belgium) connected to Chronojump System 0.9.3 (Chronojump Boscosystem, Barcelona, Spain). Gates were spaced 1.5 m apart and 1.3 m from the ground to avoid interference with arm motion [26]. The faster time of the two trials was used for further analysis.

#### 2.2.2. Change in Direction Speed Tests

The COD time performance test was registered with a photocell beam connected to a computer (Chronojump BoscoSystem, Barcelona, Spain) in seconds. All COD tests were performed with the players starting in a standing position, with their preferred foot forward and 0.5 m behind the first timing gate [27]. For the COD measurements, the subjects were instructed to run as fast as possible for five meters and then turn 90° or 135°, and a final 5-m sprint (Figure 1). Gates were spaced 1.5 m apart and 1.3 m from the ground. The faster time for each trial was used for further analysis.

#### 2.2.3. Single Leg Countermovement Jump Tests

The participants performed two successful jump trials with both legs in the vertical (SLCJ-V), horizontal (SLCJ-H), and lateral directions (SLCJ-L) [9]. Participants were trained to stand on one leg with their hands on their hips, descend into a countermovement of self-selected depth, and then quickly extend the stance leg to jump as high as possible. They were not allowed to swing the opposite leg prior to the jump. In addition, they were also instructed to land on both feet simultaneously. A trial was considered successful if balance was maintained for at least three seconds after landing. In all tests, the higher distance between the two trials was used for the analysis. The SLCJ-V height was calculated from flight time [28] with a contact mat system (Chronojump Boscosystem, Barcelona, Spain). The best time for each task was used for statistical analysis.

#### 2.2.4. Flywheel Resistance Skills

Participants performed four specific skills (acceleration step, deceleration step, crossover step, and sidestep) (Figure 2) with a flywheel resistance (iso-inertial) device (Eccotek Training Force^®^-Byomedicsystem, Barcelona, Spain). The device has a metal flywheel (diameter: 0.17 m) with up to 18 weights (0.45 kg and 0.05 m diameter each one), which involve force production during both phases of muscle contraction (concentric and eccentric). The flywheel has an axis fixed in the center. The weight rotates around this axis. The moments of inertia were 0.10 kg/m^2^ and 0.29 kg/m^2^ for the 4 and 18 weights, respectively. The moment of inertia was modified by adding weights to the flywheel and/or by selecting one of four positions (P1, P2, P3, or P4), which moves the pulley [19]. Participants completed six familiarization sessions with each flywheel resistance skill before data collection. The training of skills, such as changes of direction through flywheel devices, implies a high coordination difficulty [22,23]. To ensure the correct technique and use of the eccentric overload that this kind of technology allows, a long period of familiarization is required.

On each testing day, participants were instructed to perform a progressive test to determine the maximum power produced (based on familiarization results) of eight repetitions on each limb at maximum effort, starting with the weaker limb. The mean power of the best four repetitions of each set was recorded using a rotatory axis encoder (Chronojump Bosco-System, Barcelona, Spain) and associated Chronojump software (v. 1.8.1-95-gaebf429). The test started with 12 weights (based on familiarization results), and 2 weights were added after every set if the power of the preceding set was exceeded. This procedure was repeated until all 18 weights were added. As previously described [19], the P1 position was utilized across all tests in this sample.

The rest period between each set was two minutes. Concentric (CON) action was defined as the acceleration phase, while eccentric (ECC) action was defined as the deceleration phase. Participants were positioned one meter from the conical pulley and were encouraged to perform the task at maximum effort on familiarization and testing days. They were also instructed to generate force eccentrically throughout the specific skill and to correctly execute the action. 

### 2.3. Statistical Analysis

Means and standard deviations (SD) were calculated for all data. The Shapiro–Wilk test was used to assess the normality of the data. Moreover, the within-session reliability of test measures was analyzed using a two-way random intraclass correlation coefficient (ICC) with an absolute agreement (95% confidence intervals) and coefficient of variation (CV). Intraclass correlation coefficient (ICC) values were defined as follows: >0.9 = excellent, 0.75–0.9 = good, 0.5–0.75 = moderate, and <0.5 = poor [29]. CV values were considered acceptable at <10% [30].

Kappa coefficients (κ) were calculated to classify agreement with regard to how consistently the direction of asymmetry favored the same side [7]. Kappa values were interpreted in line with suggestions from Viera and Garret [31] and considered as follows: ≤0 = poor, 0.01–0.20 = slight, 0.21–0.40 = fair, 0.41–0.60 = moderate, 0.61–0.80 = substantial, and 0.81–0.99 = almost perfect [32]. 

To identify asymmetry between limbs, the asymmetry index (ASI) was calculated using the following formula [33,34]:$$\mathrm{ASI}\%=\frac{\mathrm{Highest}\;\mathrm{Performing}\;\mathrm{Limb}\;-\;\mathrm{Lowest}\;\mathrm{Performing}\;}{\mathrm{Highest}\;\mathrm{Performing}\;\mathrm{Limb}}\times 100$$

The Highest Performing Limb (HPL) was defined as the side with the higher value for each task, while the Lowest Performing Limb (LPL) was defined as the side with the lower value for each task. To calculate COD deficit time, the 10 m sprint time was subtracted from the COD time in each direction and for each leg [27]:COD deficit time=COD time−10 m sprint time

To identify differences between limbs, paired sample Wilcoxon tests were used to compare HPL and LPL. Cohen’s *d* Effect Sizes (ES) were used to quantify the magnitude of the difference between HPL and LPL [35]. According to Hopkins et al. [36], the values were interpreted as follows: <0.20 = trivial, 0.20–0.60 = small, 0.61–1.20 = moderate, 1.21–2.0 = large, and >2.0 = very large, following Hopkins et al. [36]. Spearman’s correlations (*ρ*) were used to compare ASI scores with physical performance tests. Statistical significance was established at *p* ≤ 0.05. The magnitude of the correlation was evaluated and interpreted as follows: trivial (0.00–0.09), small (0.10–0.29), moderate (0.30–0.49), large (0.50–0.69), very large (0.70–0.89), nearly perfect (0.90–0.99), and perfect (1.00) [36].

## 3. Results

Table 1 shows descriptive statistics for both limb data, the percentage of asymmetry for each task, and reliability measures for all assessments. Most assessments had excellent within-session ICC values (≥0.9).

The results showed that there was a significant difference between legs across all tasks (*p* < 0.05). The mean ASI values ranged from 1.26% (COD 135°) to 11.75% (SLCJ-V).

Tests only showed significant relationships in inter-limb asymmetry scores between sidestep eccentric and COD 90° plus SLCJ-L. In addition, Kappa coefficients and descriptive agreement showed that asymmetries rarely favored the same side between tests (Kappa = −0.05 to 0.58), showing different directionality depending on the test. Figure 3 and Figure 4 display the individual asymmetries for each test. In these figures, positive values indicated a right limb becoming advantage, and negative values indicated a left limb becoming advantage.

Correlations (*ρ*) between inter-limb asymmetry and test scores are shown in Table 2 and Table 3. The results only showed a significant correlation between the asymmetry of sidestep CON with SLCJ-L performance from the HPL (*ρ* = −0.61 (−0.02 to −0.88); *p* = 0.046) and between the asymmetry of sidestep CON with the different COD deficit performance measures (COD deficit 135° HPL: *ρ* = 0.78 (0.34 to 0.94); *p* = 0.005; COD deficit 135° LPL: *ρ* = 0.72 (0.21 to 0.92); *p* = 0.013; COD deficit 90° HPL: *ρ* = 0.81 (0.4 to 0.95); *p* = 0.003; COD deficit 90° LPL: *ρ* = 0.74 (0.26 to 0.93); *p* = 0.009). 

## 4. Discussion

The aims of this study were to measure inter-limb asymmetries using jumping, COD, and flywheel resistance skills tests in a sample of elite youth female basketball players and to determine their relationship to athletic performance. The main finding was that the magnitude of asymmetry varied across tests, with the SLCJ-V demonstrating the greatest magnitude of asymmetry. Moreover, the ASI and favored side differed among tests. The results also showed that higher inter-limb asymmetries during sidestep CON were related to reduced COD deficit and SLCJ-L performance.

To the best of our knowledge, this is the first study reporting the power values of specific inertial skills (acceleration step, deceleration step, sidestep, and crossover step) in elite youth female basketball athletes. The first finding of this study was the significant difference between HPL and LPL in all tests. Differences between limbs are commonly found in many team sport athletes and can be explained by sports demands, which can result in an increased frequency of unilateral movements. Athletes commonly favor one limb over the other, facilitating better coordination and increased strength in the more commonly used limb [37]. Unilateral movements in basketball include jumps for rebounding, COD, asymmetric movement with dribbling, and passing/throwing actions [38]. In line with previous research, the magnitude of inter-limb asymmetry varied by test, which likely resulted in the ASI varying by task. This finding demonstrates the importance of obtaining a battery of measurements and of not relying on a single test to quantify inter-limb differences. This will provide a complete picture of athletes’ asymmetries. It is also important to emphasize that the mean eccentric power was greater than the mean concentric power in all flywheel tests. This finding confirms that the players were experienced with flywheel resistance technology. Training with flywheel systems requires a high degree of familiarization to allow athletes to develop greater eccentric power relative to concentric power [39]. Flywheel resistance training is useful in team sports because eccentric actions during changes of direction, decelerations, and landings play important roles in both performance enhancement and injury prevention.

Regarding the inter-limb asymmetry values, the highest average ASI was identified in the SLCJ-V test (11.75 ± 7.79%), which is similar to findings from the existing literature on team sport athletes [5,11]. Flywheel power test asymmetries were lower in magnitude compared to asymmetries in the SLCJ-V (ranging from 10.86 ± 4.37 to 4.59 ± 2.43), but higher in magnitude compared to asymmetries during COD (1.26 ± 0.94 for COD 135° and 1.60 ± 1.50 for COD 90°). COD tests showed the lowest magnitude of asymmetry. This could be because this task has a strong linear speed component, which masks the existing asymmetry during COD tests [19]. Moreover, sprint tests are more reliable than jump power tests [40] and flywheel resistance tests [21]. Apart from the differences in magnitude, the HPL differed for each test for several players, which indicates that there is heterogeneity among tests in asymmetry directionality. The Kappa coefficient, which was computed to determine the consistency of asymmetry agreement across tests, showed only ‘fair’ levels of agreement (−0.36 to 0.23). This was similar to previous research from Madruga-Parera et al. [41], who identified minimal agreement for side consistency among jump tests (−0.05 to 0.15), or research from Bishop et al. [15], who identified minimal agreement for side consistency in peak force in the SLC-J and single-leg broad jump. As such, these results emphasize the heterogeneity of asymmetry among tests and highlight the need for a more individual approach to data analysis [7,41,42]. 

For physical performance, the only significant associations identified were between inter-limb asymmetries during sidestep CON, and COD deficit and SLCJ-L performance. Although there is no existing research on basketball players, other evidence has shown conflicting findings across team sports. Recently, Madruga-Parera, Bishop, Beato, et al. [20] found significant associations between magnitude of asymmetry in a crossover exercise with a flywheel resistance device and 20-m sprint time (r = 0.46) and bilateral COD performance for 90° (r = 0.48–0.51) and 180° (r = 0.41–0.51) in a sample of youth male handball players. In contrast, Raya-González et al. [21] did not find any relationships between asymmetry values using a flywheel resistance test (lateral squat) and any athletic performance score (vertical jump height tests, sprint times, or COD times in 90°) in a sample of U15 elite male soccer players. Similarly, Madruga-Parera, Bishop, Fort-Vanmeerhaeghe, et al. [19], failed to find significant associations between asymmetry values with flywheel resistance tests (lateral step and crossover step with COD) and unilateral jump performance in elite youth tennis players.

Interestingly, none of the aforementioned studies analyzed the relationship between inter-limb asymmetries and COD deficit performance. In our study, the only significant correlations reported were with COD deficit performance. Because this study did not include any mechanistic investigation, such as technical or kinematic analysis of COD ability, it is not possible to fully explain the relationships identified. However, this relationship is probably due to the fact that movements share very similar motor patterns: sudden lateral braking action (ECC) combined with rear exit (CON).

While these findings are useful, there are some limitations that should be addressed in further research. First, this study utilized a cross-sectional design, which did not allow for the establishment of causal relationships. As such, the results of the present research only represent the point in time the measurements occurred (post-season); the results may vary depending on season timing. Furthermore, it would be interesting to analyze the relationship between inter-limb asymmetries and external load (i.e., distance covered, amount of high-speed running, etc.) to understand the impact of inter-limb asymmetries on basketball practice and competition performance. Additionally, future studies could make comparisons using individual baseline data and roles on the team, expanding the sample and categorizing findings by position (i.e., point guard, small forward, center, etc.). Finally, this study also did not include kinematic analysis, as all testing was field-based. Laboratory-based studies can provide more detailed technical information, such as entry and exit velocities, when athletes jump or change direction [43,44]. Therefore, it would be interesting to carry out additional laboratory-based data collection that could quantify the mechanical components of these skills in basketball players.

## 5. Conclusions

In conclusion, this study presents a set of useful tests to identify inter-limb asymmetries in female youth basketball players. Among these, it is worth highlighting the use of flywheel devices as a novel method for assessing asymmetries. The results of this set of tests indicate that mean asymmetry magnitudes greatly varied among all metrics and showed different directionality, thus highlighting the lack of consistency across the tests and the task-specific nature of inter-limb asymmetry. 

Due to the relationship between certain inter-limb asymmetries and poorer performance, it is suggested that strength and conditioning training interventions consider reducing the magnitude of inter-limb asymmetries in female youth basketball players. It is also suggested to conduct a battery of fitness tests to provide a complete picture of inter-limb asymmetries (such as jumps, changes of direction, or power-specific skills). Finally, the findings from the current study suggest that flywheel resistance devices could be included in the battery of tests to detect inter-limb asymmetries. Since the use of this type of tool is increasingly common in elite training, it can be used to calculate asymmetries in specific exercises that athletes usually perform in their sports modality.

## Figures and Tables

**Figure 1 biology-11-00812-f001:**
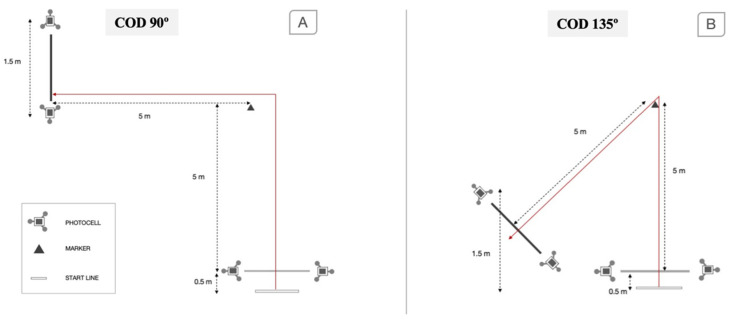
Schematic representation of the change direction tests, 90° (**A**) and 135° (**B**).

**Figure 2 biology-11-00812-f002:**
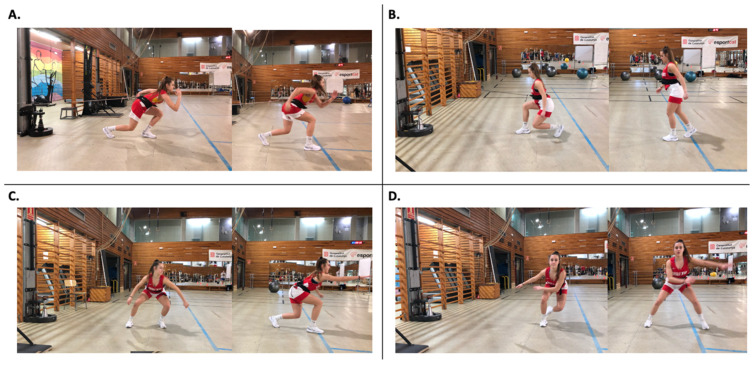
Testing conducted on the flywheel resistance device. (**A**) Acceleration step, (**B**) Deceleration step, (**C**) Crossover step, (**D**) Side step.

**Figure 3 biology-11-00812-f003:**
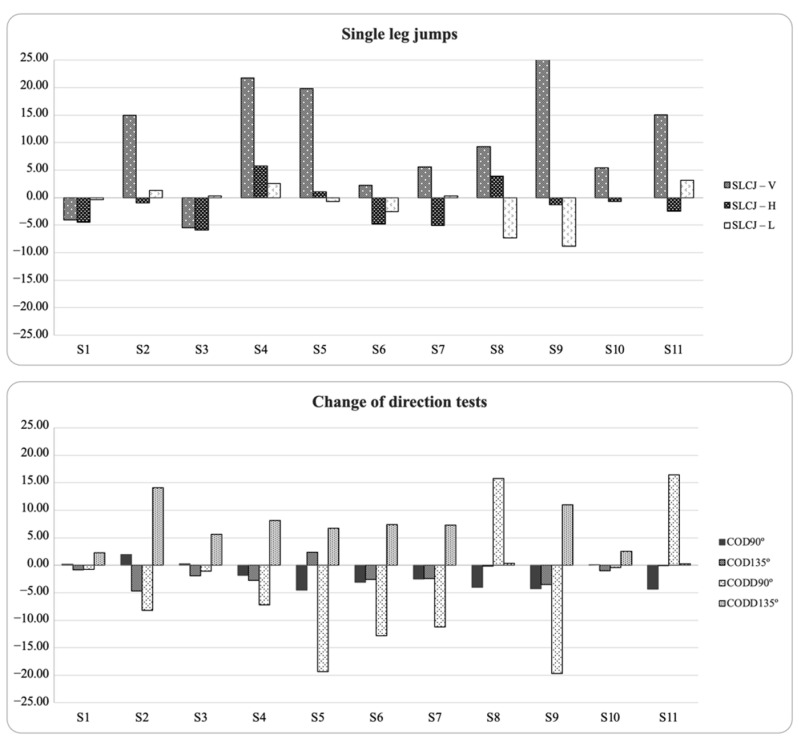
Percentage of asymmetry index (ASI) for each participant and task (positive = right leg dominance; negative = left leg dominance).

**Figure 4 biology-11-00812-f004:**
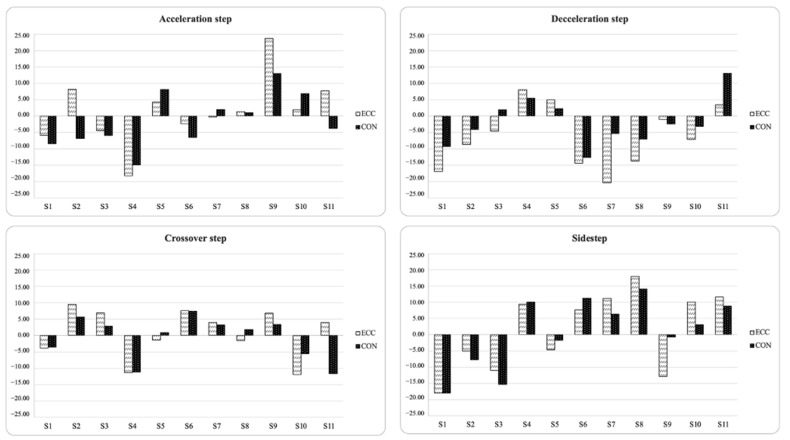
Percentage of asymmetry index (ASI) for each participant and task (positive = right leg dominance; negative = left leg dominance).

**Table 1 biology-11-00812-t001:** Mean test scores, effect sizes, inter-limb asymmetry values, and test reliability data.

Test		Mean ± SD	*p*	ES	Asymmetry (%)	ICC (95% CI)	CV (%)
Acceleration step-CON (w)	HPL	615.35 ± 138.19	0.008	0.31	6.36 ± 5.27	0.98 (0.95–0.99)	22.46
	LPL	574.85 ± 126.13				0.98 (0.96–0.99)	21.94
Acceleration step-ECC (w)	HPL	639.07 ± 135.30	0.005	0.34	6.24 ± 5.33	0.97 (0.93–0.99)	21.17
	LPL	596.97 ± 119.75				0.98 (0.95–0.99)	20.06
Crossover step-CON (w)	HPL	558.60 ± 87.23	0.001	0.34	4.79 ± 2.92	0.97 (0.93–0.99)	15.62
	LPL	530.87 ± 75.96				0.98 (0.95–0.99)	14.31
Crossover step-ECC (w)	HPL	601.76 ± 80.27	0.001	0.52	6.6 ± 4.00	0.96 (0.90–0.99)	13.34
	LPL	561.58 ± 74.06				0.99 (0.97–0.99)	13.19
Deceleration step-CON (w)	HPL	462.28 ± 52.30	0.002	0.46	5.95 ± 4.17	0.97 (0.92–0.99)	11.31
	LPL	435.77 ± 60.56				0.98 (0.94–0.99)	13.90
Deceleration step-ECC (w)	HPL	522.15 ± 61.48	0.001	0.67	8.81 ± 6.61	0.94 (0.84–0.98)	11.77
	LPL	476.99 ± 72.09				0.96 (0.89–0.99)	15.11
Sidestep-CON (w)	HPL	467.01 ± 79.55	0.000	0.47	8.84 ± 5.63	0.98 (0.95–0.99)	17.03
	LPL	427.42 ± 88.43				0.98 (0.95–0.99)	20.69
Sidestep-ECC (w)	HPL	508.06 ± 100.57	0.000	0.59	10.86 ± 4.37	0.98 (0.95–0.99)	19.79
	LPL	452.28 ± 87.72				0.98 (0.95–0.99)	19.40
SLCJ-V (m)	HPL	0.16 ± 0.03	0.002	0.67	11.75 ± 7.79	0.97 (0.91–0.99)	8.37
	LPL	0.14 ± 0.03				0.97 (0.90–0.99)	9.68
SLCJ-H (m)	HPL	1.58 ± 0.11	0.017	0.42	2.45 ± 2.75	0.81 (0.29–0.95)	6.96
	LPL	1.54 ± 0.08				0.86 (0.35–0.96)	5.19
SLCJ-L (m)	HPL	1.53 ± 0.09	0.011	0.44	2.61 ± 2.80	0.80 (0.31–0.95)	5.88
	LPL	1.49 ± 0.09				0.80 (0.05–0.93)	6.04
COD 135° (s)	HPL	3.02 ± 0.15	0.001	0.27	1.26 ± 0.94	0.93 (0.87–0.97)	4.97
	LPL	3.06 ± 0.15				0.96 (0.85–0.99)	4.90
COD 90° (s)	HPL	2.60 ± 0.11	0.006	0.40	1.60 ± 1.50	0.92 (0.72–0.98)	4.23
	LPL	2.65 ± 0.14				0.95 (0.89–0.99)	5.28
COD deficit 135° (s)	HPL	1.06 ± 0.10	0.001	0.40	6.53 ± 6.01	0.97 (0.95–0.98)	9.43
	LPL	1.10 ± 0.10				0.97 (0.94–0.99)	9.09
COD deficit 90° (s)	HPL	0.64 ± 0.06	0.006	0.71	3.62 ± 2.74	0.95 (0.91–0.97)	9.38
	LPL	0.69 ± 0.08				0.96 (0.92–0.98)	11.59
10 m sprint (s)		1.96 ± 0.08					4.08

Key: ECC = Eccentric phase; CON = Concentric phase; SLCJ-V = Vertical countermovement jump; SLCJ-H = Horizontal countermovement jump; SLCJ-L = Lateral countermovement jump; COD = Change direction capacity; HPL = Highest performing limb; LPL = Lowest performing limb; ES: Effect size; ICC = intraclass correlation coefficient; CI = confidence intervals; CV = coefficient of variation.

**Table 2 biology-11-00812-t002:** Spearman’s r correlations between inter-limb asymmetry scores and unilateral jump and COD performance.

% Asymmetry	SLCJ-V		SLCJ-H		SLCJ-L		COD 135		COD 90		COD Deficit 135		COD Deficit 90	
	HPL	LPL	HPL	LPL	HPL	LPL	HPL	LPL	HPL	LPL	HPL	LPL	HPL	LPL
Acceleration step-CON	0.12	−0.33	−0.41	−0.37	−0.09	−0.17	0.46	0.47	0.52	0.38	−0.06	−0.03	−0.17	0.00
Acceleration step-ECC	−0.36	−0.43	−0.17	−0.16	0.20	0.00	0.30	0.30	0.48	0.26	0.15	0.04	0.32	0.24
Crossover step-CON	0.11	0.07	0.20	0.48	0.11	0.38	−0.12	−0.13	0.01	−0.04	0.17	0.19	0.36	0.26
Crossover step-ECC	0.22	0.17	0.04	0.06	−0.13	0.02	−0.44	−0.49	−0.42	−0.53	−0.44	−0.58	−0.45	−0.36
Decceleration step-CON	0.06	−0.07	0.20	−0.23	0.42	0.27	0.29	0.36	0.29	0.32	0.25	0.27	0.14	0.29
Decceleration step-ECC	−0.33	−0.25	0.09	−0.73	0.30	0.04	−0.12	−0.12	−0.27	−0.10	−0.07	−0.12	−0.24	−0.24
Sidestep-CON	−0.34	−0.12	−0.52	−0.32	**−0.61 ***	−0.60	0.45	0.43	0.45	0.45	**0.78 ****	**0.72 ***	**0.81 ****	**0.74 ****
Sidestep-ECC	−0.37	−0.44	−0.10	0.14	0.11	−0.26	0.11	0.15	0.06	0.26	0.17	0.23	0.16	0.29

Key: ECC = Eccentric phase; CON = Concentric phase; SLCJ-V = Vertical countermovement jump; SLCJ-H = Horizontal countermovement jump; SLCJ-L = Lateral countermovement jump; COD = Change direction capacity; HPL = Highest performing limb; LPL = Lowest performing limb. * (*p* < 0.05); ** (*p* < 0.01).

**Table 3 biology-11-00812-t003:** Spearman’s r correlations between inter-limb asymmetry scores and flywheel resistance performance.

% Asymmetry	Acceleration Step-CON		Acceleration Step-ECC		Crossover Step-CON		Crossover Step-ECC		Decceleration Step-CON		Decceleration Step-ECC		Sidestep-CON		Sidestep-ECC	
	HPL	LPL	HPL	LPL	HPL	LPL	HPL	LPL	HPL	LPL	HPL	LPL	HPL	LPL	HPL	LPL
SLCJ-V	0.22	0.16	0.22	0.18	0.49	0.33	0.28	0.26	0.35	0.33	0.20	0.38	0.50	0.56	0.44	0.55
SLCJ-H	0.19	0.16	0.15	0.16	0.04	0.26	0.02	0.00	0.26	0.18	0.22	0.17	−0.25	−0.26	−0.11	−0.16
SLCJ-L	−0.26	−0.15	−0.29	−0.47	−0.18	−0.18	−0.19	−0.07	−0.40	−0.52	0.51	−0.62	0.15	0.07	0.28	0.14
COD 135°	−0.36	−0.34	−0.29	−0.02	−0.23	−0.30	−0.36	−0.11	−0.30	−0.18	0.24	0.07	−0.02	0.00	−0.02	0.09
COD 90°	0.02	0.10	−0.02	−0.16	0.19	0.03	0.06	0.19	−0.17	−0.26	−0.37	−0.37	0.29	0.24	0.25	0.19
COD deficit 135°	−0.31	−0.30	−0.23	0.06	−0.19	−0.26	−0.31	−0.06	−0.26	−0.11	0.31	0.16	0.01	0.28	−0.01	0.13
COD deficit 90°	−0.02	0.05	−0.06	−0.17	0.18	0.01	0.05	0.16	−0.18	−0.26	−0.36	−0.35	0.26	0.22	0.19	0.17

Key: ECC = Eccentric phase; CON = Concentric phase; SLCJ-V = Vertical countermovement jump; SLCJ-H = Horizontal countermovement jump; SLCJ-L = Lateral countermovement jump; COD = Change direction capacity; HPL = Highest performing limb; LPL = Lowest performing limb.

## Data Availability

The data presented in this study are available on reasonable request from the corresponding author.

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
