# Peer review of "Relationship between Performance and Inter-Limb Asymmetries Using Flywheel Resistance Device in Elite Youth Female Basketball Players"

_biology, 2022, doi:10.3390/biology11060812_

Round 1

Reviewer 1 Report

This study determined the magnitude and directionality of inter-limb asymmetries from jumping, COD, and flywheel resistance skills tests and examine the relationship of asymmetries to physical performance in youth elite female basketball players.

This is an interesting and pertinent study. The design, methods, statistical analysis used were appropriate and the results original to warrant publication. The results were appropriately presented, and the discussion was based on sound reasoning.

The submitted manuscript has many qualities, оverall, the presented data were interesting. Although I applaud the author(s) on attempting to obtain data, I have minor comments for the authors.

- Introduction:  Well-written overall.

- Materials and Methods: 

Lines 137-138 and 148-149 – Please explain the need for the stated time and number of test attempts conducted in order to familiarize with the given tests and procedures

Line 154 - 2.2.110. m Linear Sprint possible typo?

Lines 184, 191, 265, 267, Table 1, Figure 3… - Abbreviations for jump tests were presented different through the text, it needs to be unified.

Lines 194 to 224 - At first reading Flywheel resistance skills test does not have a clear explanation. Please clarify.

Line 245 – Please add citations for COD deficit time formula. 

- Results:

Table 1 – different font through the table, it needs to be unified.

Figures 3 and 4 - Are they really necessary in the article? They burden the article, while the results that are presented in them not sufficiently explained through the text.

- Discussion:  Well-written overall.

- Conclusions:

Need to be more specific with the conclusions; why and what were relationships established?

Also, through the all text please check the authors guidelines required by the Journal. Especially according to Abbreviations and SI Units, it needs to be unified.

Author Response

Rebuttal letter

Please note that:

[R1] = comments from Reviewer #1.

[A] = answers from the authors.

REVIEWER 1

[R1] This study determined the magnitude and directionality of inter-limb asymmetries from jumping, COD, and flywheel resistance skills tests and examine the relationship of asymmetries to physical performance in youth elite female basketball players.

This is an interesting and pertinent study. The design, methods, statistical analysis used were appropriate and the results original to warrant publication. The results were appropriately presented, and the discussion was based on sound reasoning.

The submitted manuscript has many qualities, оverall, the presented data were interesting. Although I applaud the author(s) on attempting to obtain data, I have minor comments for the authors.

[A] Thanks very much for the reviewer’s comments and suggestions. All of them have significantly improved the quality of the text.

[R1] Materials and Methods: 

Lines 137-138 and 148-149 – Please explain the need for the stated time and number of test attempts conducted in order to familiarize with the given tests and procedures

[A] We have rewritten the “Design and Procedures” section in order to clarify the protocol used. “One week before data collection, participants were familiarized with all tests and procedures. Participants were tested on five separate days, each separated by 48 hours, and the training was always on the same schedule in order to control the circadian rhythms. In the first four testing days, the following flywheel skills data were collected in order: 1) acceleration step, 2) deceleration step, 3) sidestep, and 4) crossover step. Day 5 consisted of unilateral jump, 10 m linear sprint and COD tests. During testing days, all participants completed the same standardized warm-up consisting of 7 min of cardiovascular exercise (RPE 5-6), 7-minutes of multidirectional displacements, 3-minutes of dynamic stretching exercises (e.g. walking lunges, high knee lifts, side steps) and 3-minutes of maximal and progressive intensity displacements including changes of direction, jumps, and acceleration/deceleration movements. The warm-up was supervised by a qualified strength and conditioning coach and consistent feedback was provided throughout all tests to ensure proper technique. On completion, 3 practice trials were provided for each test where subjects were instructed to perform them at 75, 90, and 100% of their perceived maximal effort. Three minutes of rest was given between the last practice trial and the start of the first test. Both the order of the tests and the participants were randomized with the “true random number generator” program.

Line 154 - 2.2.110. m Linear Sprint possible typo?

[A] Thanks. We have corrected

Lines 184, 191, 265, 267, Table 1, Figure 3… - Abbreviations for jump tests were presented different through the text, it needs to be unified.

[A] Thanks for the comment. We have unified the abbreviations.

Lines 194 to 224 - At first reading Flywheel resistance skills test does not have a clear explanation. Please clarify.

[A] We have rewritten this paragraph to improve the clarity of the explanation. “Participants performed four specific skills (acceleration step, deceleration step, crossover step and sidestep) (Figure 2) with a flywheel resistance (iso-inertial) device (Eccotek Training Force® - Byomedicsystem, Barcelona, Spain). The device has a metal flywheel (diameter: 0.17 m) with up to 18 weights (0.45 kg and 0.05 m diameter each one) which involve force production during both phases of muscle contraction (concentric and eccentric). A fixed axis is located at the center of the beam around which the weights rotate. The moments of inertia were 0.10 kg_m2 and 0.29 kg_m2 for 4 and 18 weights, respectively. The modification of the moment of inertia was made by adding any of the 18 weights to the edge of the flywheel and also by selecting one of four positions (P1, P2, P3 or P4), which changes the location of the pulley [19]. Participants completed six familiarization sessions with each flywheel resistance skill before data collection.

Line 245 – Please add citations for COD deficit time formula. 

[A] Done:

Nimphius, S.; Callaghan, S.J.; Spiteri, T.; Lockie, R.G. Change of Direction Deficit: A more isolated measure of change of direction performance than total 505 time. J. Strength Cond. Res. 2016, 30, 3024–3032, doi:10.1519/JSC.0000000000001421.

[R1] Results:

Table 1 – different font through the table, it needs to be unified.

[A] Done

Figures 3 and 4 - Are they really necessary in the article? They burden the article, while the results that are presented in them not sufficiently explained through the text.

[A] We think that they are necessary, since they show the directionality of the asymmetries in each participant for each test. While looking at the tables, only the magnitude of that asymmetry can be discerned. However if the reviewer considers, it can be deleted.

- Conclusions:

Need to be more specific with the conclusions; why and what were relationships established?

[A] We rewrite the whole paragraph to clarify the comprehension and to be more specific.

In conclusion, the current study presents a set of useful tests to identify inter-limb asymmetries in female youth basketball players. Among these, it is worth highlighting the use of flywheel devices as a new tool to assess asymmetries. The results of this set of tests indicate that mean asymmetry magnitudes greatly varied among all metrics and showed different directionality, thus highlighting the lack of consistency across the tests and the task-specific nature of inter-limb asymmetry.

Due to the association between some inter-limb asymmetries and reduced performance, it is suggested that strength and conditioning training interventions should consider the reduction of inter-limb asymmetry in female youth basketball players. It is also suggested to conduct a battery of fitness tests in order to provide a holistic picture of inter-limb asymmetries (such as jumps, changes of direction or power specific skills). Finally, the findings from the current study suggest that the use of flywheel resistance devices could be included in the battery of tests for the detection of inter-limb asymmetries. Since the use of this type of tool is increasingly common in elite training, it could be used to calculate asymmetries in specific exercises that athletes usually perform in their sports modality.”

Also, through the all text please check the authors guidelines required by the Journal. Especially according to Abbreviations and SI Units, it needs to be unified.

 [A] We have checked the journal’s style and made several corrections. Thanks

Reviewer 2 Report

My opinion is that overall merit of the paper is to low for this journal. Since the work is in the field of sports, maybe a journal from that field would be more appropriate. 

At the first sight this is an interesting and relevant study for the area of sport, and I commend the authors on the data collected. I have minor comments for the authors:

Introduction:

Introduction is well-written overall. Perhaps you can add hypothesis but it’s not required.

Line 60 and 61 – I would use singular instead of plural  (’’the game’’ instead of ’’games’’; ’’ space ’’instead of ’’spaces’’).

Line 105 – maybe ’’relationship’’ is more appropriate term than ’’associations’’.

Line 112 - maybe ’’using’’ is more appropriate term than ’’from’’.

Materials and Methods:

Line 117 – I have count 10 tests. Maybe I was wrong.

Line 123 – Do you have enough power with 11 subjects?

Line 124 - Specify instruments for obtaining anthropometric measures.

Line 128-130 – I would use sentence ’’They usually have 7-9 training sessions per week plus one game during the weekend’’.

Line 137-153 – On what day a 10m sprint test was performed? And please can you explain why they need so many test attempts for familiarization and re-familiarization? Different abbreviations for the jump tests are used throughout the text , they need to be unified .

Line 154 – missing dot after 2.2.1? Please explain how do you get HPD (or HPL) and LPD (or LPL) for the COD test?

Lines 194 to 224 – As we can see from the figures, Flywheel resistance skills test is done with arm swing. Can that affect the test result as lower extremity asymmetry is required? Please provide more detailed information.

Results:

The results are presented appropriately, with few minor comments:

Line 262 – How you can explain slightly lower values of ICC in SLCJ-L and SLCJ-H in relation to SLCJ-V?

Table 1 - The abbreviations in the table and in the key of the table are different (HPD and LPD; HPL and LPL)

Figure 4 – Equate the negative y-axis of the crossover step with other ones.

Discussion:  Well-written overall.

Line 303 – I would use ’’using SLCJ-V......’’ instead of ’’from a battery of fitness tests’’.

Line 313, 357, 358, 361 – ’’a sample of’’ is unnecessary.

Line 325 and 326 – ’’in this sample of youth female athletes’’ is unnecessary.

Line 383, 384 – ’’kinematic’’ instead of ’’technical’’; ’’tests were’’ instead of ’’testing was’’

Conclusions:

Need to be more specific with the conclusions; why and what were relationships established?

At the end:

After re-reading, I can change my decision to Reconsider after major revision...but I still think that significance of the main claims are low. They are novel, still not convincing enough. Kinetic and kinematic analysis can strengthen the paper.

Author Response

Rebuttal letter

Please note that:

[R2] = comments from Reviewer #2.

[A] = answers from the authors.

REVIEWER 2

[R2] My opinion is that overall merit of the paper is to low for this journal. Since the work is in the field of sports, maybe a journal from that field would be more appropriate. 

At the first sight this is an interesting and relevant study for the area of sport, and I commend the authors on the data collected. I have minor comments for the authors:

[A] We really appreciate the reviewer for it time and dedication to review this manuscript. We have reviewed and considered his/her recommendations and corrections which have significantly improved the quality of the text.

It is true that the present journal has a different scope. However, the authors think that the topic of this special issue entitled “Biological, Physiological, and Biomechanical Determinants of Human Performance Optimization” is appropriate for the submitted article. In this sense, some of the topics covered by this special issue are: sports performance, neuromuscular performance, strength and conditioning, performance testing, etc.

Introduction:

Introduction is well-written overall. Perhaps you can add hypothesis but it’s not required.

Line 60 and 61 – I would use singular instead of plural (’’the game’’ instead of ’’games’’; ’’ space ’’instead of ’’spaces’’).

[A] Changed.

Line 105 – maybe ’’relationship’’ is more appropriate term than ’’associations’’.

[A] Done. Thanks.

Line 112 - maybe ’’using’’ is more appropriate term than ’’from’’.

[A] Changed.

Materials and Methods:

Line 117 – I have count 10 tests. Maybe I was wrong.

[A] Yes, it was a mistake, they are ten. Thanks.

Line 123 – Do you have enough power with 11 subjects?

[A] We totally agree with the reviewer’s point of view. This is a limitation of the study that must be acknowledged. However, it must be taken into account that the sample used is young elite female basketball players, which participate in a talent development program. Moreover, most of them (90 %) are members of the Spanish national team. In this sense, it is difficult to carry out studies with a large sample of these specific characteristics.

Line 124 - Specify instruments for obtaining anthropometric measures.

[A] Anthropometrical measures were reported by the medical staff, using the conventional instruments following the International Society for the Advancement Kineanthropometry (ISAK) protocol. If the reviewer considers it, it can be specified in the text.

Line 128-130 – I would use sentence ’’They usually have 7-9 training sessions per week plus one game during the weekend’’.

[A] Changed. ‘All participants were involved in a four-year talent development program at the time of the study. They usually have 7-9 training sessions per week plus one game during the weekend

Line 137-153 – On what day a 10m sprint test was performed? And please can you explain why they need so many test attempts for familiarization and re-familiarization?

Thanks for the comment, it was a mistake. We have added in the text. In the first four testing days, the following flywheel skills data were collected in order: 1) acceleration step, 2) deceleration step, 3) sidestep, and 4) crossover step. Day 5 consisted of unilateral jump, 10 m linear sprint and COD tests.”

The training of skills such as changes of direction through flywheel devices implies a high coordination difficulty (see: Nuñez Sanchez & Sáez de Villarreal, 2017; Raya-González et al., 2021). In order to ensure a correct technique and use of the eccentric overload that this kind of technology allows, a long period of familiarization is required. This is one of the reasons why there are so few studies using inertial training, especially with female population. It requires a long period of familiarization to be able to acquire adaptations.

Nuñez Sanchez, F. J., & Sáez de Villarreal, E. (2017). Does Flywheel Paradigm Training Improve Muscle Volume and Force? A Meta-Analysis. Journal of Strength and Conditioning Research, 31(11), 3177–3186. https://doi.org/10.1519/JSC.0000000000002095

Raya-González, J., de Keijzer, K. L., Bishop, C., & Beato, M. (2021). Effects of flywheel training on strength-related variables in female populations. A systematic review. Research in Sports Medicine, 11(1), 1–18. https://doi.org/10.1080/15438627.2020.1870977

Different abbreviations for the jump tests are used throughout the text, they need to be unified .

[A] Thanks for the comment. We have unified the abbreviations.

Line 154 – missing dot after 2.2.1? Please explain how do you get HPD (or HPL) and LPD (or LPL) for the COD test?

[A] Thanks. It has been corrected.

We also have changed the HPD for HPL (Highest Performance Limb) in order to maintain the same acronym throughout the manuscript. The HPL is defined as the limb with the best performance in each specific test since most times could be different depending on the test. In the case of the COD test, the HPL is the limb with which the change of direction is made, and a shorter time is achieved in the test (and therefore better performance). In the case of jump tests or tests with the inertial device, the limb with which a greater jump distance or greater power is achieved.

See:

Fort-Vanmeerhaeghe, A., Montalvo, A. M., Sitjà-Rabert, M., Kiefer, A. W., & Myer, G. D. (2015). Neuromuscular asymmetries in the lower limbs of elite female youth basketball players and the application of the skillful limb model of comparison. Physical Therapy in Sport, 16(4), 317–323. https://doi.org/10.1016/j.ptsp.2015.01.003

Lines 194 to 224 – As we can see from the figures, Flywheel resistance skills test is done with arm swing. Can that affect the test result as lower extremity asymmetry is required? Please provide more detailed information.

[A] We agree with the reviewer's point of view. Flywheel resistance skills test are less analytical than, for example, the unilateral jump tests (hands on hips). However, this is a positive aspect, since the flywheel resistance skills test tries to reproduce the specific sports movements (which are executed as holistic movements).

Results:

The results are presented appropriately, with few minor comments:

Line 262 – How you can explain slightly lower values of ICC in SLCJ-L and SLCJ-H in relation to SLCJ-V?

[A] This is a common fact in different studies. It is probably due to the fact that the SLCJ-V has a simpler technical component than the other tests. In the SLCJ-L or SLCJ-H, the reception made by the athlete is more difficult since it moves from the site and this generates a greater challenge of dynamic stability.

Table 1 - The abbreviations in the table and in the key of the table are different (HPD and LPD; HPL and LPL)

[A] Corrected

Figure 4 – Equate the negative y-axis of the crossover step with other ones.

[A] Done. Thanks.

Discussion: Well-written overall.

Line 303 – I would use ’’using SLCJ-V......’’ instead of ’’from a battery of fitness tests’’.

[A] Done. “The aims of this study were to measure inter-limb asymmetries using jumping, COD, and flywheel resistance skills tests, in a sample of youth elite female basketball players and to determine their relationship to athletic performance”

Line 313, 357, 358, 361 – ’’a sample of’’ is unnecessary.

[A] Deleted. “To the best of our knowledge, this is the first study reporting the power values of specific inertial skills (acceleration step, deceleration step, sidestep, and crossover step) in elite youth female basketball athletes.”

Line 325 and 326 – ’’in this sample of youth female athletes’’ is unnecessary.

[A] Deleted. “This will provide a holistic picture of asymmetries in athletes. It’s also important to emphasize that the mean eccentric power was greater than the mean concentric power in all flywheel tests.”

Line 383, 384 – ’’kinematic’’ instead of ’’technical’’; ’’tests were’’ instead of ’’testing was’’

[A] Changed. Finally, this study also did not include kinematic analysis as all testing were field-based.”

Conclusions:

Need to be more specific with the conclusions; why and what were relationships established?

[A] We rewrite the whole paragraph to clarify the comprehension and to be more specific.

In conclusion, the current study presents a set of useful tests to identify inter-limb asymmetries in female youth basketball players. Among these, it is worth highlighting the use of flywheel devices as a new tool to assess asymmetries. The results of this set of tests indicate that mean asymmetry magnitudes greatly varied among all metrics and showed different directionality, thus highlighting the lack of consistency across the tests and the task-specific nature of inter-limb asymmetry.

Due to the association between some inter-limb asymmetries and reduced performance, it is suggested that strength and conditioning training interventions should consider the reduction of inter-limb asymmetry in female youth basketball players. It is also suggested to conduct a battery of fitness tests in order to provide a holistic picture of inter-limb asymmetries (such as jumps, changes of direction or power specific skills).Finally, the findings from the current study suggest that the use of flywheel resistance devices could be included in the battery of tests for the detection of inter-limb asymmetries. Since the use of this type of tool is increasingly common in elite training, it could be used to calculate asymmetries in specific exercises that athletes usually perform in their sports modality.”

At the end:

After re-reading, I can change my decision to Reconsider after major revision...but I still think that significance of the main claims are low. They are novel, still not convincing enough. Kinetic and kinematic analysis can strengthen the paper.

[A] Thanks very much for the reviewer’s comments and suggestions. All of them have significantly improved the quality of the text.

We agree that with a kinetic and kinematic analysis the paper would have more impact. In fact, this is one of the main limitations that the authors acknowledge. Finally, this study also did not include kinematic analysis as all testing were field-based. Laboratory-based studies can provide more detailed technical information, such as entry and exit velocities when athletes jump or change direction [42,43]. Therefore, it would be interesting to carry out kinematic analysis that could help to assess the mechanical components of these skills in basketball players.”

However, one strength of the present paper lies in the novelty of the instruments since few studies have reported in a sample of elite female athletes, the lower limb imbalances using flywheel devices. A tool that is increasingly used in the field of strength training in professional sports.  The evaluation with flywheel devices (with an encoder) or with contact mats gives instantaneous feedback of athletes' results, which in high performance is very useful. Something that does not happen with kinetic and kinematics analysis, which requires a high economic cost and time to analyze data.

Reviewer 3 Report

lines 141-3- justify order of the tests

lines 162-70- sustain with rehability test references

lines 263-5- paired t test is not mentioned in statistical analysis, and statistical value was not included in Table 1. Additionally, if Spearman was used, it is assumed that no normal distribution exist (but no information was given about that), meaning that parametric comparison tests are not recommended (isn't it?)

lines 284-9- include confidence intervals

Author Response

Rebuttal letter

Please note that:

[R3] = comments from Reviewer #3.

[A] = answers from the authors.

REVIEWER 3

[A] We really appreciate the reviewer for it time and dedication to review this manuscript. We have reviewed and considered his/her recommendations and corrections which have significantly improved the quality of the text. We have further developed the conclusion section according to the three reviewers’ suggestions.

 [R3]  lines 141-3- justify order of the tests

[A] The order of the test was also randomized with the “true random number generator” program.

lines 162-70- sustain with rehability test references

[A] Done.

lines 263-5- paired t test is not mentioned in statistical analysis, and statistical value was not included in Table 1. Additionally, if Spearman was used, it is assumed that no normal distribution exist (but no information was given about that), meaning that parametric comparison tests are not recommended (isn't it?)

[A] We have included in statistical analysis section. “To identify differences between limbs, the magnitude of differences between HPL and LPL were assessed with paired sample t-tests. The magnitude of the difference was determined using Cohen’s d Effect Sizes (ES) [35]. Values were interpreted as: < 0.20 = trivial, 0.20-0.60 = small, 0.61-1.20 = moderate, 1.21-2.0 = large and > 2.0 = very large, following Hopkins et al. [36].”

Yes it is. As some test values do not have normal distribution (“The Shapiro-Wilk test was used to check the normality of the tested parameters.”), Spearman was used (Spearman’s correlations (ρ) were used to compare ASI scores with physical performance tests.”).

Round 2

Reviewer 2 Report

Dear authors,

It looks much better after revision, congratulations on your work and on your reply...Perhaps you could add some comments from your reply in the manuscript (at the end of the Line 213, for example: The training of skills such as changes of direction through flywheel devices implies a high coordination difficulty (see: Nuñez Sanchez & Sáez de Villarreal, 2017; Raya-González et al., 2021). In order to ensure a correct technique and use of the eccentric overload that this kind of technology allows, a long period of familiarization is required), but it’s not required...

Author Response

Thanks for the suggestion. It has been added in the text.

Reviewer 3 Report

Study still needs to prove and strenghten some statistical details, in order to sustain discussion and conclusions and to achive Q2 level, namely, paired t test requirements were not and must be proved and Spearman confidence intervals must be presented. The problem is: if data have no normal distribution, paired t test can not be applied, and, due to sample size Spearman value must be accompanied by confidence intervals. 

Author Response

Rebuttal letter

Please note that:

[R3] = comments from Reviewer #3.

[A] = answers from the authors.

REVIEWER 3

[R3] Study still needs to prove and strenghten some statistical details, in order to sustain discussion and conclusions and to achive Q2 level, namely, paired t test requirements were not and must be proved and Spearman confidence intervals must be presented. The problem is: if data have no normal distribution, paired t test can not be applied, and, due to sample size Spearman value must be accompanied by confidence intervals. 

[A] Thank you for the correction. In the first report, we did not realize that we had mistranslated the explanation of the statistical test used.

“To identify differences between limbs, paired sample Wilcoxon tests were used to compare HPL and LPL. Cohen’s d Effect Sizes (ES) was used to quantify the magnitude of the difference between HPL and LPL [35].”

Regarding the correlations’ confidence intervals, they have been added for the significant correlations. If the reviewer considers it necessary, the non-significant correlations can also be added, although Tables 2 and 3 will be very extensive.

"Correlations (ρ) between inter-limb asymmetry and test scores are shown in Table 2 and Table 3. Results only showed a significant correlation between the asymmetry of sidestep CON with SLCJ-L performance from the HPL (ρ = -0.61 (-0.02 to -0.88); p = 0.046) and between the asymmetry of sidestep CON with the different COD deficit performance measures (COD deficit 135º HPL: ρ = 0.78 (0.34 to 0.94); p = 0.005; COD deficit 135º LPL: ρ = 0.72 (0.21 to 0.92); p = 0.013; COD deficit 90º HPL: ρ = 0.81 (0.4 to 0.95); p = 0.003; COD deficit 90º LPL: ρ = 0.74 (0.26 to 0.93); p = 0.009)."

We really appreciate the reviewer for his time and dedication to reviewing this manuscript. Their comments have significantly improved the methodological quality of the text.